# EndoGeneAnalyzer: A tool for selection and validation of reference genes

**Eliel Barbosa Teixeira**[ID][1]*, **André Salim Khayat**[1], **Paulo Pimentel Assumpção**[1], **Samir Mansour Casseb**[ID][1], **Caroline Aquino Moreira-Nunes**[ID][1,2,3], **Fabiano Cordeiro Moreira**[1]

**1** Oncology Research Center, Federal University of Pará, Belém, Pará, Brazil, **2** Pharmacogenetics Laboratory, Drug Research and Development Center (NPDM), Department of Medicine, Federal University of Ceará, Fortaleza, Brazil, **3** Clementino Fraga Group, Central Unity, Molecular Biology Laboratory, Fortaleza, CE, Brazil

* eliel.teixeira@icen.ufpa.br

## Abstract

The selection of proper reference genes is critical for accurate gene expression analysis in all fields of biological and medical research, mainly because there are many distinctions between different tissues and specimens. Given this variability, even in known classic reference genes, demands of a comprehensive analysis platform is needed to identify the most suitable genes for each study. For this purpose, we present an analysis tool for assisting in decision-making in the analysis of reverse transcription-quantitative polymerase chain reaction (RT-qPCR) data. EndoGeneAnalyzer, an open-source web tool for reference gene analysis in RT-qPCR studies, was used to compare the groups/conditions under investigation. This interactive application offers an easy-to-use interface that allows efficient exploration of datasets. Through statistical and stability analyses, EndoGeneAnalyzer assists in the select of the most appropriate reference gene or set of genes for each condition. It also allows researchers to identify and remove unwanted outliers. Moreover, EndoGeneAnalyzer provides a graphical interface to compare the evaluated groups, providing a visually informative differential analysis.

## Introduction

Reverse transcription-quantitative polymerase chain reaction (RT-qPCR or qPCR) is a highly sensitive and specific technique used to study gene expression in many research fields, such as human disease, because of its capacity to detect rare transcripts and observe small variations in gene expression [1–4].

RT-qPCR is a technique widely used for quantifying gene expression levels. By quantifying the the RNA molecules present in a sample, RT-qPCR provides valuable insights into the expression of specific genes. To ensure accurate and reliable results, reference genes are used in the normalization process. These reference genes are stably expressed under various experimental conditions and serve as internal controls to normalize the gene expression data. Normalization with reference genes allows for a more accurate comparison of gene expression levels across different samples or experimental conditions, eliminating potential variations

**Data Availability Statement:** This study presents the EndoGeneAnalyzer tool, available at https://npobioinfo.shinyapps.io/endogeneanalyzer/ and the

open-source code can be found at https://github.com/MoreiraFC/EndoGeneAnalyzer.

**Funding:** This study was supported by Brazilian funding agencies: Coordination for the Improvement of Higher Education Personnel (CAPES; to E.B.T), the National Council of Technological and Scientific Development (CNPq grant number [404213/2021-9 to CAM-N; Productivity in Research PQ scholarships to P.P.A, A.S.K., and CAM-N]), and the Cearense Foundation of Scientific and Technological Support (FUNCAP grant number [P20-0171-00078.01.00/20 to CAM-N]); we also thank PROPESP/UFPA for the publication payment. There was no additional external funding received for this study.

**Competing interests:** The authors declare no conflict of interest. The funders had no role in the study's design; in the collection, analyses, or data interpretation; in the writing of the manuscript, or in the decision to publish the results.

caused by factors such as RNA quality, sample quantity, or technical variations [5]. According to Chervoneva et al [6], among the essential criteria for choosing a good reference gene are: the level of expression unaffected by experimental factors, minimal variability in its expression between tissues and physiological states of the organism, and, preferably, that the gene has a quantification cycle (Cq) value similar that of the target gene. The Cq value indicates the position of the amplification curve with respect to the cycle axis [7].

According to the MIQE guidelines, the selection and the number of reference genes are essential, especially because they need to be experimentally validated for each specific sample type and study condition [8, 9]. Thus, ideal reference genes should have a minimum intersubject variation in terms of quantification cycle (Cq) values, and it is recommended that this variation of the reference genes between samples be less than 1 Cq [8]. This characteristic is crucial in the data normalization process for expression comparisons, as it ensures accurate mRNA concentration measurements and reliable conclusions [10]. The normalization of data using reference genes involves correcting errors that arise from the initial concentration of RNA/cDNA, and the most common method used in RT-qPCR assays involves the use of one or more reference genes [11].

Studies have demonstrated the variability of commonly used reference genes, such as *GAPDH*, *β2M*, and *18S*, under various conditions [12–15]. In aging studies, variability in *GAPDH* expression, when used as an internal control, interferes with the detection of subtle variations in the target genes under investigation [16]. Selecting an unstable reference gene for qPCR normalization can compromise experimental accuracy. Commonly used reference genes such as *18S*, *ACTB*, and *GAPDH* may not always be suitable for this purpose and should be validated for stability in a specific study context [11]. Choosing an inappropriate reference gene leads to inaccurate normalization and misleading conclusions. This approach may also introduce variability and bias, hindering comparisons between samples [10].

In this context, algorithms have been developed to help identify the most appropriate reference genes from a given set of candidate genes. NormFinder [17], geNorm [18], BestKeeper [19], RefGenes [9], and RefFinder [20] are some software tools available for this purpose, with RefFinder being the only web-based tool currently available.

In this study, we developed the EndoGeneAnalyzer tool, available at https://npobioinfo.shinyapps.io/endogeneanalyzer/. The open-source code can be found at https://github.com/MoreiraFC/EndoGeneAnalyzer. This tool is a dynamic web-based tool for comparing and selecting the most stable set of reference genes from a dataset derived from RT-qPCR experiments. It also integrates NormFinder software [17]. Unlike existing algorithms, this tool allows the identification of variations by group/condition and the removal of outliers present in reference genes, a step often overlooked in most gene expression studies. Furthermore, the tool provides ability to analyze the differential expression of the target genes across different groups/conditions, allowing the investigation of differences in the expression of the gene of interest and the identification of potential associations with experimental conditions.

## Materials and methods

### EndoGeneAnalyzer platform

EndoGeneAnalyzer is a dynamic web-based platform that simplifies and assists in selecting reference genes in scientific studies and performing differential gene expression analysis for RT-qPCR data. With interactive interfaces and a statistical approach, the tool facilitates the identification of the best reference genes for the investigated groups or conditions. The EndoGeneAnalyzer workflow is illustrated in Fig 1.

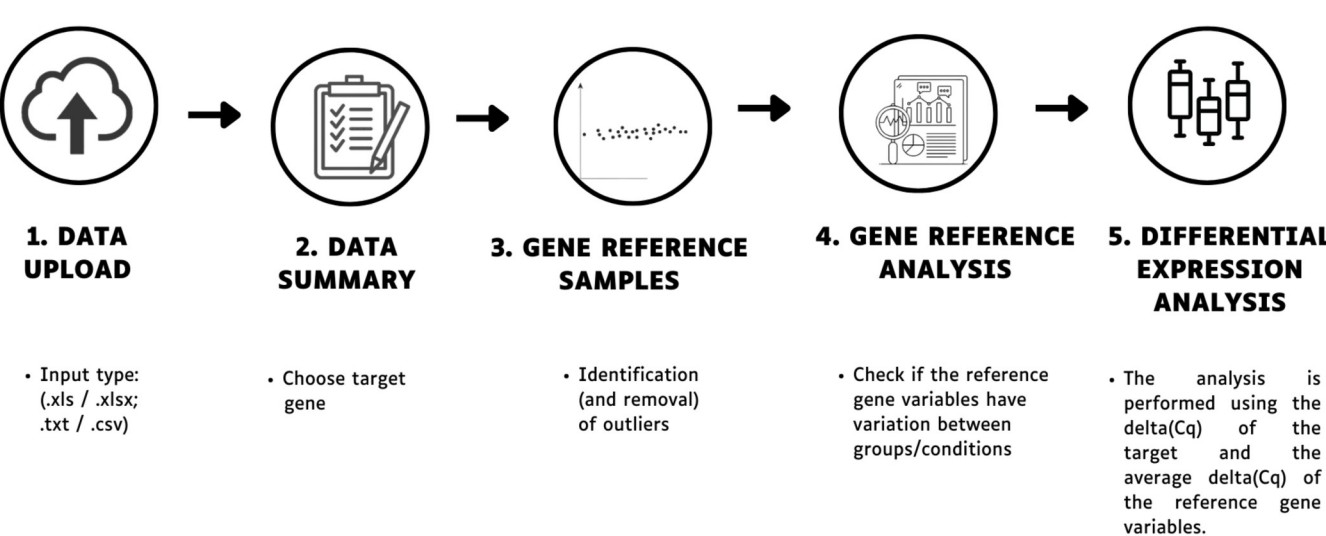

**Fig 1. EndoGeneAnalyzer tool workflow.** Users import data and select target genes for analysis, and the tool allows outlier removal. It also calculates stability metrics are calculated, the best reference gene is identified, and differential expression analysis between groups or conditions is enabled.

The tool has been developed to be intuitive, interactive, and efficient, with several steps to guide the user. In the first step, the user enters the data with the option to choose between the supported file formats:.xls/.xlsx or.txt/.csv. This flexibility makes it easy to import data from different sources. After loading the data, the user selects the targets of interest for analysis (nonreference genes). The next step focuses on evaluating the best reference set of genes based on mean variation using descriptive statistical data such as gene standard deviation, the sum of squared differences between the mean of each group and the gene mean, and the sum of squared differences between the standard deviation of each group and the gene standard deviation. Stability metrics are also calculated using NormFinder, which helps to identify the genes that best fit the study conditions. One of the critical innovations of EndoGeneAnalyzer is the ability to analyze the stability of reference genes. In this step, the tool allows the user to identify and remove outliers, which are samples with ΔCq mean values above or below a user-defined threshold (default = 2 standard deviations), providing flexibility in the analysis.

Finally, the EndoGeneAnalyzer can perform differential expression analysis using the target ΔCq and the mean ΔCq of the set of reference genes. This step allows accurate and efficient comparisons between different groups or conditions, further delivering a fold change result.

## Data upload

This step is critical to ensure that correct information is used during the analysis. The input file must contain the following columns: i) the first column with the sample names; ii) the following columns with the mean Ct values of specific targets and reference genes for the sample; and iii) the last column with information about the groups or conditions to which the samples belong.

The data can be imported in two ways: i) the Excel tables (.xls/.xlsx) option, in which the tool does not require modification of the decimal separator; and ii) text tables (.txt/.csv)

option, in which the default decimal separator is dot(.) and in which it is necessary to configure the text delimiter.

Finally, after verifying the correct formatting of the table, the user needs to click on the "Confirm Data Table" button to proceed with the analysis process. This final step ensures that the tool correctly recognizes and validates the data provided.

## Data summary

The selection of target genes (nonreference) is essential because it is crucial to guide the analysis; these genes are related to the research objectives. Once the target-gene(s) have been identified, the user must click the "Update Target Gene" button to confirm the selection. This step ensures that the selected genes are processed and included in the analysis.

## Gene reference samples

Outliers are atypical data values that can be identified in RT-qPCR data, with experimental errors being the leading cause of these occurrences. These errors are related to environmental conditions, instrument calibration problems, or other sources of uncontrolled variation that may occur during the experiment [21]. It is essential to be aware of outliers and to understand the potential impact of their removal on results.

EndoGeneAnalyzer identifies outliers per group for each gene and their removal can be easily performed using an available function. By default, the tool considers a sample as an outlier if the mean $\Delta Cq > |2|$ standard deviation from the mean of the group/condition to which the sample belongs for the reference gene. This value can be configured according to the user's preferences. The tool offers two methods of outlier removal: i) removal of all outliers and ii) removal of those that directly interfere with the mean Cq values of the reference genes.

The user can choose which outlier to show and remove by clicking on the "Choose which outlier to remove" radio button. "Only Mean" will show and remove only outliers in the mean of the reference genes. "All Outliers" will show and remove outliers in each gene individually. This second option tends to result in the removal of more outliers and, consequently, more samples from the analysis. Removing outliers is an interactive process since it decreases the group standard deviation and may reveal additional outliers. In addition, the tool's dynamic interface facilitates the restoration of outliers as part of the analysis process.

## Gene reference analysis

This is a crucial step in the tool's operation, as it provides information about the reference genes and their variation between the different groups or conditions studied. Significant changes in the reference genes are observed at this stage, especially in the mean values between the analyzed groups.

The first table generated is the "Gene Reference by group", which presents information about the variation observed between the groups or conditions studied for each reference gene or the averages of the group of reference genes. The statistical tests used are the Wilcoxon-Mann-Whitney test (2 groups) or Kruskall-Wallis/Dunn test (3 or more groups). At this stage, it is expected that there will be no significant changes (p-value > 0.05) in reference genes between the studied groups or conditions.

The tool also provides the "Gene Reference Descriptive Statistics" table, which presents three fundamental values for assessing the reference genes: the gene standard deviation, the sum of squared differences between the mean of each group and the gene mean, and the sum of squared differences between the standard deviation of each group and the gene standard deviation. The formulas used to calculate the sum of squared differences are as follows: $n$ is the

number of groups, $\mu_i$ is the mean Cq of the group, $\mu_g$ is the mean Cq of the gene, $\sigma_i$ is the standard deviation of the group and $\sigma_g$ is the standard deviation of the gene.

$$sum.mean.square.diff = \sum_{i=0}^{n} (\mu_i - \mu_g)^2$$

$$sum.SD.square.diff = \sum_{i=0}^{n} (\sigma_i - \sigma_g)^2$$

In addition, NormFinder provides information on the stability and suitability of the reference, since this software is integrated into our tool interface. The NormFinder software employs an ANOVA-based model to account for intra- and intergroup variations. The generated table consists of four defined columns: the first column is the gene name, the second column (GroupDif) represents a measure of the difference between the groups, the third column (GroupSD) is the common standard deviation within a group, and the fourth column (Stability) provides the stability value. Thus, genes with lower stability values exhibit less variable expression and maintain a consistently stable expression pattern, while genes with higher stability values exhibit variable expression and uphold a less stable expression pattern [17].

## Differential expression analysis

EndoGeneAnalyzer allows for comparison gene expression differences among the investigated groups using ΔCq. ΔCq was calculated as the difference between the target gene and the mean of the reference genes. For a given 2 groups, two statistical tests are integrated into the tool: the Pearson t test and the Wilcoxon-Mann-Whitney Rank Sum test. For the comparisons between 3 or more groups, ANOVA/Tukey's teste and Kruskall-Wallis/Dunn test are applied.

The system also calculates the Shapiro test for the normality of each group and the Fold-Change using the formula 2-ΔΔCq [22]. This metric quantifies the difference in expression between a given two groups, considering the relative variation in ΔCq values.

## Development of the EndoGeneAnalyzer web tool

The EndoGeneAnalyzer tool was developed with the Shiny framework in R studio software (v.1.7.4, https://shiny.rstudio.com/) [23] that transforms regular R code into an interactive environment that can follow and "react" to remote user instructions. The tool is compatible with all commonly used internet browsers.

The interactive visualizations and tables were rendered using the ggplot2 (v.3.4.2, https://ggplot2.tidyverse.org), knitr (v1.42, https://cran.r-project.org/web/packages/knitr), and DT (v.0.19, https://CRAN.R-project.org/package=DT) R packages. Kruskal-Wallis and Dunn's tests were performed using the dunn.test R package (https://cran.r-project.org/web/packages/dunn.test/), and all the other statistical tests were performed using R basic statisitcs.

## Results

To illustrate the usefulness of EndoGeneAnalyzer, we employed unpublished RT-qPCR data from our laboratory, which can be accessed on the "Tutorial" tab at the following web address: https://npobioinfo.shinyapps.io/endogeneanalyzer/. After confirming the table submission, the data were loaded, and the first analysis panel titled "Data Summary" will be available. In this panel, the loaded Cq averages were displayed in a graph highlighting the conditions specified during the upload (Fig 2), which initially allowed us to observe of the dynamics of the reference and target genes under the studied conditions. The user also selects the target gene for further analysis at this stage.

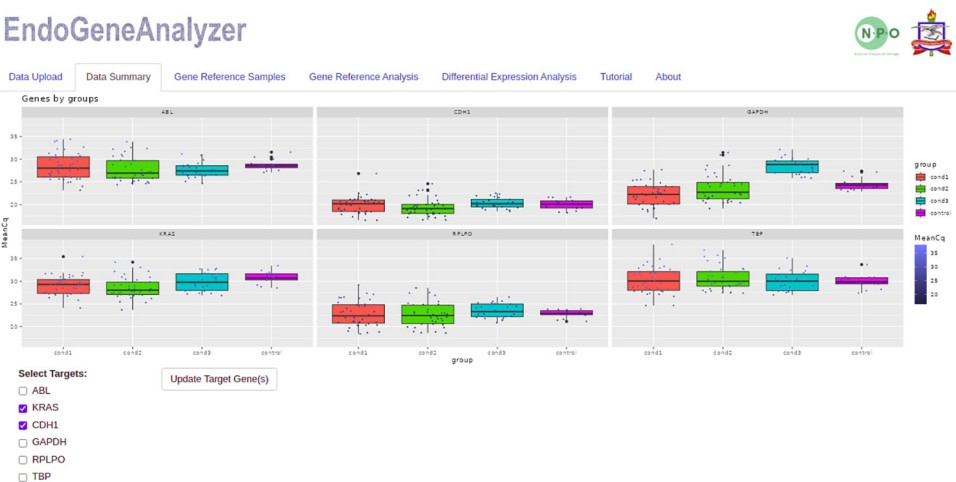

**Fig 2. Analysis of specific reference genes.** This figure provides a comprehensive side-by-side analysis of specific reference genes. Each gene is visually represented by a distinct square, allowing for a clear and concise depiction of its individual attributes. Within each square, box plots showcase the mean Cq values for each condition examined in the study. The implementation of vivid colors throughout the figure serves to effectively differentiate the diverse groups or conditions under investigation, enriching our comprehension of their unique characteristics and trends.

Subsequently, as previously mentioned regarding outliers and their impact on RT-qPCR data analysis, EndoGeneAnalyzer allows handling these values to be handled. In the "Gene Reference Samples" panel, the user can identify and remove these values in two ways: i) remove "All outliers" or ii) remove those outliers that affect the reference gene averages "Only mean", as shown in Fig 3.

Furthermore, in the "Gene Reference Analysis" tab, statistical reports are generated for the selected reference genes based on ΔCt and the mean of the reference genes. These reports provide information that supports the choice of the most stable set of reference genes (Fig 4). The first report is the result of nonparametric tests, with the choice of the test being conditional on

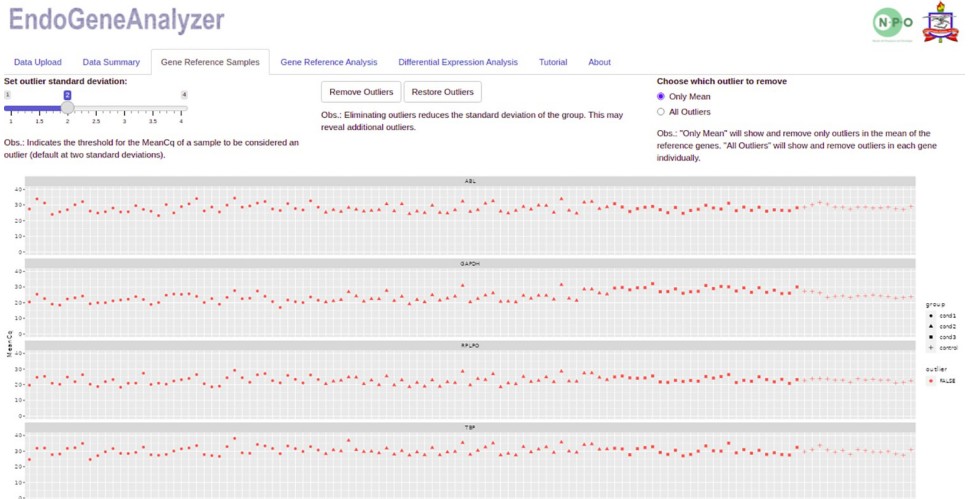

**Fig 3. Graphical verification of potential outliers separated by analyzed groups.** This figure shows four rectangles, each representing the sample distribution for a single reference gene. The icons (dot, triangle, square, and cross) symbolize the conditions, with red indicating non-outlier data and blue icons indicating outlier data. At the top, the standard deviation can be adjusted, and the user can remove outliers that affect the mean or all outliers.

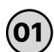

## EndoGeneAnalyzer

Data Upload  Data Summary  Gene Reference Samples  **Gene Reference Analysis**  Differential Expression Analysis  Tutorial  About

**01 Gene Reference by group**

The first line of the table is the p-value of the Kruskall-Wallis test for each reference gene.
The other lines are p-values of the Wilcoxon-Mann-Whitney test between each condition/group.
The last column is the tests p-value of the reference genes mean.
Red values indicate p-values < 0.05. It is desirable that reference genes and/or the reference genes mean do not vary significantly among groups/conditions.

Search:

|  | ABL | GAPDH | RPLPO | TBP | MeanRef |
|---|---|---|---|---|---|
| **Kruskal-Wallis** | **0.177** | **0.000** | **0.368** | **0.915** | **0.026** |
| cond1 - cond2 | 0.180 | 0.043 | 0.475 | 0.684 | 0.379 |
| cond1 - cond3 | 0.211 | 0.000 | 0.176 | 0.495 | 0.015 |
| cond2 - cond3 | 0.397 | 0.000 | 0.321 | 0.841 | 0.017 |
| cond1 - control | 0.236 | 0.006 | 0.412 | 0.580 | 0.250 |
| cond2 - control | 0.135 | 0.099 | 0.491 | 1.000 | 0.216 |
| cond3 - control | 0.144 | 0.002 | 0.393 | 0.463 | 0.192 |

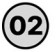

**02 Gene Reference Descriptive Statistics**

It is preferable for reference genes not to vary across different conditions or groups.
Additionally, it is desirable for reference genes to exhibit low variances,
enhancing the ability to discern small yet significant differences in the target.
If a reference gene exhibits high standard deviation and/or has low stability and/or is significantly altered across group/condition, consider to remove it from the analysis.

|  | ABL | GAPDH | RPLPO | TBP | MeanRef |
|---|---|---|---|---|---|
| Standard.Deviation | 2.42 | 3.35 | 2.36 | 2.47 | 2.28 |
| sum.SD.square.diff | 2.45 | 7.40 | 3.17 | 1.22 | 2.03 |
| sum.mean.square.diff | 0.89 | 22.50 | 0.39 | 0.29 | 1.32 |

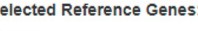

**Selected Reference Genes:**
☑ ABL
☑ GAPDH
☑ RPLPO
☑ TBP

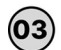

**03 NormFinder Analysis**

NormFinder Article

NormFinder is a method to evaluate reference gene stability using the variation of the gene expression.
The lower the stability value, the more stable the gene is considered.

NormFinder Result Ordered

|  | GroupDif | GroupSD | Stability |
|---|---|---|---|
| RPLPO | 0.83 | 1.01 | 0.49 |
| TBP | 2.11 | 0.96 | 0.90 |
| ABL | 2.87 | 1.15 | 0.98 |
| GAPDH | 5.74 | 1.27 | 1.71 |

NormFinder Result Unordered by group

|  | GroupDif | GroupSD | Stability | IGroupSD.V1 | IGroupSD.V2 | IGroupSD.V3 | IGroupSD.V4 | IGroupDif.V1 | IGroupDif.V2 | IGroupDif.V3 | IGroupDif.V4 |
|---|---|---|---|---|---|---|---|---|---|---|---|
| ABL | 2.87 | 1.15 | 0.98 | 1.53 | 1.14 | 0.64 | 0.44 | 0.90 | -0.07 | -1.43 | 0.61 |
| GAPDH | 5.74 | 1.27 | 1.71 | 1.77 | 0.91 | 0.70 | 1.17 | -1.93 | -0.76 | 2.87 | -0.18 |
| RPLPO | 0.83 | 1.01 | 0.49 | 1.27 | 0.83 | 1.03 | 0.48 | 0.41 | 0.11 | -0.38 | -0.14 |
| TBP | 2.11 | 0.96 | 0.90 | 1.17 | 0.86 | 0.87 | 0.73 | 0.62 | 0.72 | -1.05 | -0.29 |

GroupDif - Group Differences
GroupSD - Group Standard Deviation
IGroupDif - Intergroup Differences
IGroupSD - Intergroup Standard Deviation

**Fig 4. Gene Reference Descriptive Statistics by EndoGeneAnalyzer.** (1) Analysis of reference genes separated by study groups. (2) Descriptive statistics of the analyzed reference genes. (3) Results generated from the NormFinder tool. The first table displays the statistical results obtained using the Kruskal-Wallis test

for each reference gene and the mean of all genes. Statistically significant values are highlighted in red. The second table presents descriptive data for each examined group, indicating lower values for superior gene expression or a more favorable set of genes. The third table shows the stability data generated by the NormFinder software, providing insights into the reliability of the identified reference genes for accurate gene expression analysis.

the number of groups to be compared. To select the best reference genes, it is ideal that there is no significant variation among the groups, especially in the MeanRef column. The second report provides a descriptive analysis of the reference genes (standard deviation; sum.SD. square.diff and sum.mean.square.diff), with lower values indicating better reference gene(s). The last reports are additional analyses generated by the NormFinder software integrated into EndoGeneAnalyzer [16].

In the example provided, the use of the GAPDH gene alone was not a good internal control ($< 0.05$), mainly when used in combination with ABL, TBP and RPLPO under the conditions investigated (MeanRef $\leq 0.05$). However, removing GAPDH from the analysis was demonstrated to be a viable alternative for the conditions investigated (MeanRef $> 0.05$), given that it is a gene with high variability (Standard.Deviation = 3.35 and Stability = 1.71). In addition, since NormFinder analyses determine expression stability by assessing intra- and intergroup variation, it is possible to identify which reference gene varies between groups and exclude it from the analyses. RPLPO, TBP and ABL (0.49, 0.90, and 0.98, respectively) were the genes with the greatest stability; that is, they exhibited a consistently stable expression pattern, suggesting that they are excellent internal controls. The user can perform differential expression analysis in the last panel of the "Differential Expression Analysis" tool by selecting the target gene and comparison groups (Fig 5). In addition to the graphs generated based on ΔCq, fold change values, normality tests results, and statistical test tables are also displayed.

## Discussion

RT-qPCR is the gold standard for gene expression studies in molecular research and clinical practice. This method is prized for its rapidity, reproducibility, high sensitivity, and specificity,

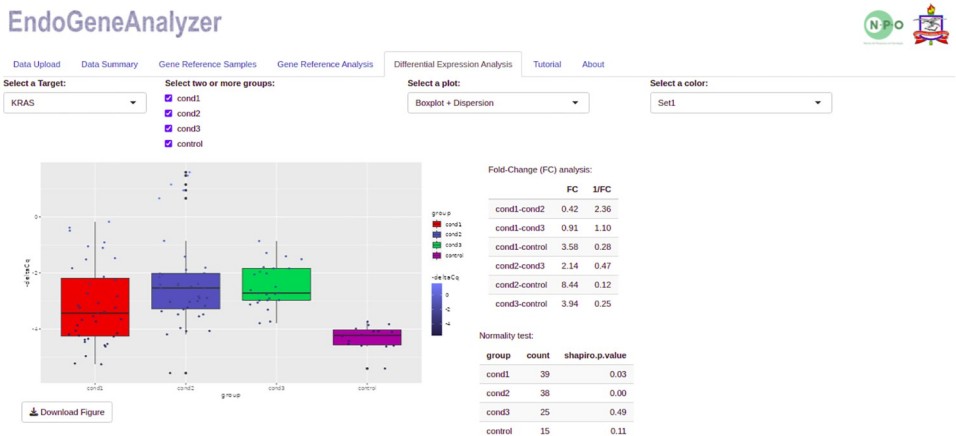

**Fig 5. Graphical visualization of differential expression analysis separated by target and study groups.** The figure summarizes the differential gene expression among the examined groups using ΔCq values in box plots. The color-coded conditions and tables showing the fold change values and Shapiro-Wilk normality test results provided comprehensive information. The user interface allows the selection of target genes and groups for statistical comparisons, including Pearson's t-test, Wilcoxon-Mann-Whitney rank sum test (for two groups), ANOVA/Tukey's test, or the Kruskal–Wallis/Dunn test (for three groups). This illustration facilitates the exploration of the molecular differences underlying gene expression variations between conditions.

enabling the detection of gene expression even in low-yield samples. Additionally, RNA-seq is recognized as an increasingly utilized and potent tool for evaluating gene expression [24]. In real-time experiments, reference genes provide a comparative basis for assessing relative variations in the expression of target genes [18]. Therefore, the identification and validation of these genes are mandatory steps.

Some studies have questioned the use of conventionally established reference genes that may fail to enable the detection of subtle differences in target gene expression under certain conditions due to their high variability [25–27] and may lead to misinterpretation of results depending on the experimental context [28].

Outlier removal is critical for the statistical analysis of qPCR data [29], as these values can bias descriptive statistics, such the as mean and standard deviation, that are used to describe gene expression. Thus, removing outliers and identifying genes that vary with the studied conditions helps to ensure accuracy and reliability in interpreting results.

EndoGeneAnalyzer is an invaluable tool for researchers conducting RT-qPCR experiments. This tool offers several features that significantly improve the accuracy and reliability of gene expression studies. It enables precise data normalization, a critical step in gene expression analysis, ensuring that the results are appropriately adjusted and comparable across samples.

## Conclusion

In summary, EndoGeneAnalyzer is a new instrument used for RT-qPCR that has produced remarkable results in terms of gene expression analysis. To address the challenge of reference genes selection, our platform, which uses simulative data, was used to demonstrate the identification and assessment suitable genes in unique study settings. All the statistical analyses were combined with stability measures and outlier filtering to allow for informed selection of reference genes to provide a stable basis for data normalization.

EndoGeneAnalyzer provides a comprehensive set of features tailored to genetic research needs, including group and variation analysis, outlier removal, user-friendly data exploration, and differential expression analysis.

EndoGeneAnalyzer promises to be a useful tool for improving the quality of gene expression experiments. Such integrations are anticipated to make the research stronger and reproducible.

## Author Contributions

**Conceptualization:** Eliel Barbosa Teixeira, Fabiano Cordeiro Moreira.

**Supervision:** Fabiano Cordeiro Moreira.

**Writing – original draft:** Eliel Barbosa Teixeira, Fabiano Cordeiro Moreira.

**Writing – review & editing:** Eliel Barbosa Teixeira, André Salim Khayat, Paulo Pimentel Assumpção, Samir Mansour Casseb, Caroline Aquino Moreira-Nunes, Fabiano Cordeiro Moreira.

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
