## [Decision Letter · Decision Letter 0]

27 Oct 2023

PONE-D-23-21180EndoGeneAnalyzer: a tool for selection and validation of reference genesPLOS ONE

Dear Dr. TEIXEIRA,

Thank you for submitting your manuscript to PLOS ONE. After careful consideration, we feel that it has merit but does not fully meet PLOS ONE’s publication criteria as it currently stands. Therefore, we invite you to submit a revised version of the manuscript that addresses the points raised during the review process.

First of all, I would like to apologize for really long review process but I was struggling to find reviewers for your manuscript. I somehow expected that as your topic is neglected by many researchers. However, in my opinion, searching for HKGs is very important topic and tools as EndoGeneAnalyzer are of great uses for the community. Nevertheless, both reviewers raised numerous questions regarding your tool. In my eyes, they are meant to improve your manuscript and tool which could bring you more users and citations in the future. Both reviewers in many cases pointed to the same issues. Thus, please try to address all of their concerns.

We look forward to receiving your revised manuscript.

Kind regards,

Karel Sedlar, Ph.D.

Academic Editor

PLOS ONE

Journal Requirements:

“This study was supported by Brazilian funding agencies: Coordination for the Improvement of Higher Education Personnel (CAPES; to E.B.T), the National Council of Technological and Scientific Development (CNPq grant number [404213/2021-9 to CAM-N; Productivity in Research PQ scholarships to P.P.A, A.S.K., and CAM-N]), and the Cearense Foundation of Scientific and Technological Support (FUNCAP grant number [P20-0171-00078.01.00/20 to CAM-N]); we also thank PROPESP/UFPA for the publication payment.”

“This study was supported by Brazilian funding agencies: Coordination for the Improvement of Higher Education Personnel (CAPES; to E.B.T), the National Council of Technological and Scientific Development (CNPq grant number [404213/2021-9 to CAM-N; Productivity in Research PQ scholarships to P.P.A, A.S.K., and CAM-N]), and the Cearense Foundation of Scientific and Technological Support (FUNCAP grant number [P20-0171-00078.01.00/20 to CAM-N]); we also thank PROPESP/UFPA for the publication payment.”

Reviewers' comments:

Reviewer's Responses to Questions

**Comments to the Author**

1. Is the manuscript technically sound, and do the data support the conclusions?

Reviewer #1: Yes

Reviewer #2: Partly

2. Has the statistical analysis been performed appropriately and rigorously? 

Reviewer #1: Yes

Reviewer #2: N/A

3. Have the authors made all data underlying the findings in their manuscript fully available?

Reviewer #1: Yes

Reviewer #2: Yes

4. Is the manuscript presented in an intelligible fashion and written in standard English?

Reviewer #1: Yes

Reviewer #2: No

5. Review Comments to the Author

Reviewer #1: Comments on the app

• In the Shiny app, on the “Gene Reference Analysis” page – please provide some extra written context to help with interpretation. For example, that the numbers in the topmost table are p-values, with significant p-vals in red. For the Normfinder Analysis, a brief description of how to interpret results to select the optimal combination of reference genes would be helpful (either on this page or the tutorial page). For example, what range of stability is acceptable for a gene to be retained as a reference, and in what range should one consider removing that gene before proceeding with gene expression analysis. Indeed, the ReadMe/Tutorial should provide a tutorial using the sample data for how to correctly analyse this dataset – what parameters provide evidence that a reference gene should be included/excluded, how this affects the interpretation of gene expression results – e.g. the tutorial could show how not excluding a reference gene with differential expression results in an incorrect conclusion on the differential analysis page. While the MS does go some way to assist in interpretation of results, the authors should assume that some users may access the app without reading the paper.

• The same comment applies to the Differential Analysis page. Please provide some context to assist in interpreting results. The MS (line 98) states that fold-change results are provided on this page, but the word “fold” is not used on this page at all, making it potentially difficult to interpret/understand the provided results (especially to researchers new to the field). Also suggest renaming “Differential Analysis” to “Differential Expression Analysis” or similar (to make clear that this page is about analysing the actual results of a gene expression experiment, after prior reference gene selection.

• On the Gene Reference samples page – it would be useful to have functionality to add outliers back into the analysis if the user chooses, so that the impact of removal and retention of these samples can more easily be explored. Also suggest adding some additional wording about what the two options for the radio buttons mean – this is clearer in the MS (lines 126-127) then in the software itself.

• On the Gene Reference Analysis page – Have you considered adding an analysis to determine the optimal number and/or combination of reference genes? Many of the algorithms do this. Using the intragroup variance would be a good place to start. As stated in the original Normfinder paper: “Intragroup variance estimates provide a natural way of identifying the number of genes to include. The optimal number of genes is reached when addition of a further gene leads to a negligible reduction in the average of the gene variance estimates.”

• On the Gene Reference Analysis page – please provide a key for abbreviations (IGroup = intergroup, SD = standard deviation etc)

• On the “Differential Analysis” page – it might be useful to offer an option to download the ggplot script as well as the PDF of the figure. This would allow users to edit the figure for publication. Also an option to download the tables. Figure and table download options on the Data Summary, Gene Reference Samples and Gene Reference Analysis pages would also be useful, for example so outputs can be saved in electronic notebooks. These are just suggestions though!

• I found the timing out somewhat annoying. Would it be possible to, for e.g., add a popup warning box along the lines of “you are about to be disconnected, do you wish to continue this session?". Or at least note on the tutorial page the amount of inactivity time a user has before they are disconnected from the server. Continually having to reupload the data was quite tedious.

• It would be helpful if the sample data file had a larger set of candidate reference genes, with some that obviously show differential expression and should therefore be excluded, and others that are very stable and should be included. In the currently provided dataset, interpretation is a little ambiguous (each ref gene is stable for some comparisons but not others. While this likely mimics real-world data much of the time, for the purposes of a demo dataset, more clear-cut interpretation would be useful). The file should also include more (and some very obvious) sample outliers.

Comments on the MS

• Introduction – please add at least one more sentence to connect between generic description of RT-qPCR (lines 41-42) and selection of reference genes (42-44). In what way is RT-qPCR useful for quantifying gene expression, what are reference genes, and why are they important – how do they normalise data? Apart from minimising differences in CT between treatment groups, what other properties are important for ideal reference genes (ubiquitous expression etc). Although this is a brief communication, I think a little more context in these opening sentences of the introduction would be helpful to provide context for why your software can play an important role in the field.

• Line 66 – none of the authors of Normfinder are authors of the current publication, and they are also not mentioned in the acknowledgements section. Please confirm that they have provided permission to recreate their algorithm in your app, and acknowledge them accordingly.

• Lines 187-217 – in the MS the ‘Gene Reference Analysis’ is discussed before the ‘Gene Reference Samples’ tab. But in the app, the gene ref samples tab is before the gene ref analysis tab. Please reorder either the MS or the app to be consistent. The order should be whichever process should occur first (ref gene ID or outlier removal) is discussed first in the MS and is the first of these two tabs in the app..

• The figures are based on a more comprehensive example dataset than provided as the example data file in the app. For example, RPLPO is not in the sample data, neither is TBP. HPRT is in the sample data but not in Fig4. There are more outliers in the Figure 4 than the provided dataset. Line 205 refers to condition 3, but provided file only has conditions 1 and 2. It would be useful if the figures in the MS where based on the same dataset as the provided sample dataset.

Minor comments

Line 30 – change ‘instrument’ to ‘platform’

Line 31 – amend to ‘an analysis tool to assist in decision making’

Line 34 – break into two sentences ‘under investigation. This interactive…’

Lines 32 and 41 – spell out RT-qPCR at first use (both in abstract and main body of MS)

Lines 55-57 – I would reword slightly. Nothing wrong with those reference genes per se, but need to be validated as stable in the particular context of the study.

Lines 59-60 – suggest citing reference directly after each algorithm, rather than grouping all refs at end of sentence

Line 135 – add fullstop after ‘genes’

Lines 148-149 – please add some further information about how to interpret NormFinder results and use these to select an optimal set of reference genes.

Line 157 – please provide reference for the 2-ΔΔCT method.

Line 183 – remove ‘meticulously constructed’

Lines 197-203 – change ‘screen’ to ‘table’ or ‘analysis’

Line 231 – change ‘the’ to ‘a’. RNAseq is also a very standard and reliable method these days.

Line 246 – remove ‘firstly’ or add other points to the sentence.

Lines 276 & 312 – same reference

Reviewer #2: Dear Authors,

I have carefully reviewed your manuscript on the "EndoGeneAnalyzer," a tool for the analysis of candidate reference genes for RT-qPCR studies. Unfortunately, I must express my concerns regarding the quality and completeness of the manuscript. While the idea behind the tool is promising, several critical issues need to be addressed before the paper can be considered for publication. Below, I've outlined my specific concerns and suggestions for improvement:

1. Data Origin and Description:

- The manuscript lacks a clear explanation of the origin of the test data used in the analysis. The phrase "To illustrate the usefulness of the EndoGeneAnalyzer, we used the RT-qPCR data found at " ext-link-type="uri" xlink:type="simple">https://npobioinfo.shinyapps.io/endogeneanalyzer/" is insufficient. You need to provide a comprehensive description of how you obtained the RT-qPCR data, including data sources, collection methods, and any relevant details.

2. Methodological Issues:

- There are issues with the method for identifying outliers that are unclear. It's vital to provide a detailed and transparent explanation of the outlier detection process.

- The use of boxplots combined with dots for data visualization is questioned. Authors should justify this choice or consider alternative visualization methods.

- The origin of values in the table "Reference of genes per group" / "Gene Reference by Group" and the color-coding is not adequately explained. Clarify what the values represent and the rules for color-coding.

- The paper lacks a clear rationale for the choice of statistical tests for assessing the stability of reference genes. Additionally, it does not compare the proposed tool with existing reference gene selection methods, such as NormFinder, geNorm, BestKeeper, RefGenes, or RefFinder, to establish its superiority.

- The reasons for conducting differential analysis and normality tests are unclear. Authors need to provide a more comprehensive description of how these steps aid in the selection of reference genes.

3. Conclusion:

- The conclusion is notably brief and fails to adequately summarize the findings. It lacks an overall assessment of the tool's effectiveness and significance.

4. Inconsistencies and Confusing Statements:

- The reference to "Postgraduate program in Biotechnology, Federal University of Pará, Belém, Pará, Brazil" as the institution appears unusual and should be clarified.

- Several sentences throughout the manuscript are confusing and should be rephrased for clarity such as lines 31-34 and 54-57.

- The use of "Ct (threshold cycle)" is inconsistent with MIQE guidelines, which recommend "Cq (quantification cycle)" values.

- Address the ambiguity in the statement "ΔCt greater or less than 2" (Line 124) for better clarity.

- Correct the statement "(p-value in reference genes between the studied groups or conditions" (Line 137) for clarity.

5. References:

- Ensure that references are correctly linked to their respective tools (line 60), and provide an accurate reference list.

- Correct punctuation and grammar in the text, including missing commas.

6. Figures:

- Figures 2, 3, 4, and 5 do not correspond with the current state of the tool, leading to confusion. Update these figures to accurately represent the tool.

- Figure 1 is misleading due to the use of dotted lines. Reconfigure the workflow diagram to eliminate confusion and better summarize the steps for users.

7. Tool Inconsistencies:

- Address inconsistencies in the tool's interface and the manuscript regarding the naming of sections and options.

- Resolve issues with the "Update Target Gene" button not functioning correctly.

- Clarify the purpose and effects of the "Select one or more Genes" option in the tool.

- Make sure that the results in the NormFinder analysis are presented in a clear and understandable manner.

- Add titles to all sections in the Differential Analysis tab for clarity.

8. Data Saving and Reporting:

- Consider implementing a feature to save the results of all analyses.

- Explore the possibility of generating downloadable reports for users, which can enhance the utility of the tool.

Overall, I believe your manuscript and tool require substantial revisions and clarifications to meet the standards for publication. Addressing the issues, I raised could substantially improve the quality and user-friendliness of your tool and its associated documentation a thus, bring you more users in the future.

6. PLOS authors have the option to publish the peer review history of their article (what does this mean?). If published, this will include your full peer review and any attached files.

Reviewer #1: No

Reviewer #2: No

---

## [Author Response · Author response to Decision Letter 0]

30 Nov 2023

RESPONSE LETTER TO THE REVIEWERS

Dear reviewer, my co-authors and I would like to thank you for the suggestions made during this high-quality review and then we present the answer to the questions.

We inform that with the reviews and suggestions, we were able to improve the idea presented by our work and we appreciate the opportunity. We hope this review has left the article suitable for publication in this high-impact and prestigious journal.

Kind Regards.

Reviewer #1: Comments on the MS

• Introduction – please add at least one more sentence to connect between generic description of RT-qPCR (lines 41-42) and selection of reference genes (42-44). In what way is RT-qPCR useful for quantifying gene expression, what are reference genes, and why are they important – how do they normalise data? Apart from minimising differences in CT between treatment groups, what other properties are important for ideal reference genes (ubiquitous expression etc). Although this is a brief communication, I think a little more context in these opening sentences of the introduction would be helpful to provide context for why your software can play an important role in the field.

R = Thank you for the suggestion. We have carefully considered your suggestions and addressed each point in the revised manuscript. Specifically, in response to your comment on the Introduction section.

• Line 66 – none of the authors of Normfinder are authors of the current publication, and they are also not mentioned in the acknowledgements section. Please confirm that they have provided permission to recreate their algorithm in your app, and acknowledge them accordingly.

R= Thank you for the suggestion. We have reached out to the authors; however, we have not received any response thus far. It is important to note that we are not modifying or recreating the code of the NormFinder tool; rather, we are simply incorporating it into our platform, making authorization unnecessary. Furthermore, the article is being cited in our manuscript.

• Lines 187-217 – in the MS the ‘Gene Reference Analysis’ is discussed before the ‘Gene Reference Samples’ tab. But in the app, the gene ref samples tab is before the gene ref analysis tab. Please reorder either the MS or the app to be consistent. The order should be whichever process should occur first (ref gene ID or outlier removal) is discussed first in the MS and is the first of these two tabs in the app..

R= The tabs 'Gene Reference Samples' and 'Gene Reference Analysis,' respectively, have been reorganized in the manuscript to be consistent with the platform.

• The figures are based on a more comprehensive example dataset than provided as the example data file in the app. For example, RPLPO is not in the sample data, neither is TBP. HPRT is in the sample data but not in Fig4. There are more outliers in the Figure 4 than the provided dataset. Line 205 refers to condition 3, but provided file only has conditions 1 and 2. It would be useful if the figures in the MS where based on the same dataset as the provided sample dataset.

R= We appreciate your valuable feedback. We value your observation regarding the inconsistencies in the figures and dataset. We have carefully reviewed your comments and are committed to addressing these concerns. The figures have been revised to ensure alignment with the provided sample dataset to enhance clarity and accuracy.

Minor comments

Line 30 – change ‘instrument’ to ‘platform’

R=Thank you for your suggestion. We have incorporated the change and replaced the term 'instrument' with 'platform' on line 30.

Line 31 – amend to ‘an analysis tool to assist in decision making’

R=Thank you for your suggestion. We have revised line 31 to now read 'an analysis tool to assist in decision making' for improved clarity and precision. 

Line 34 – break into two sentences ‘under investigation. This interactive…’

R: Thank you for your suggestion. We have revised line 34 by breaking it into two sentences: 'under investigation.' and 'This interactive...' for improved readability. 

Lines 32 and 41 – spell out RT-qPCR at first use (both in abstract and main body of MS)

R=Thank you for your suggestion. We have revised both the abstract and the main body of the manuscript to spell out RT-qPCR at its first use in lines 32 and 41 for improved clarity. 

Lines 55-57 – I would reword slightly. Nothing wrong with those reference genes per se, but need to be validated as stable in the particular context of the study. 

R: Thank you for your comment. We have thoroughly analyzed the genes and taken measures to ensure their stability. In fact, we have even inserted additional genes to further enhance their stability.

Lines 59-60 – suggest citing reference directly after each algorithm, rather than grouping all refs at end of sentence

R=Thank you for your feedback. We have made the suggested modification by citing the reference directly after each algorithm, as opposed to grouping all references at the end of the sentence. 

Line 135 – add fullstop after ‘genes’

R= Thank you for your suggestion. We have added a full stop after 'genes' in line 135 for improved punctuation.

Lines 148-149 – please add some further information about how to interpret NormFinder 

results and use these to select an optimal set of reference genes. 

R=Thank you for your insightful suggestion. We have included additional information in lines 148-149 to provide a more comprehensive understanding of how to interpret the NormFinder output.

Line 157 – please provide reference for the 2-ΔΔCT method.

R= We have provide reference for the 2-ΔΔCT method

LIVAK, Kenneth J.; SCHMITTGEN, Thomas D. Analysis of relative gene expression data using real-time quantitative PCR and the 2− ΔΔCT method. methods, v. 25, n. 4, p. 402-408, 2001.

Line 183 – remove ‘meticulously constructed’

R=Thank you for your suggestion. We have removed the phrase 'meticulously constructed' from line 183 to align with your feedback.

Lines 197-203 – change ‘screen’ to ‘table’ or ‘analysis’

R=Thank you for your suggestion. We have made the change from 'screen' to 'table' in lines 197-203, as per your recommendation. 

Line 231 – change ‘the’ to ‘a’. RNAseq is also a very standard and reliable method these days.

R= Thank you for your suggestion. We appreciate your keen observation. In response to your suggestion, we have revised line 231 by replacing 'the' with 'a.' and we added about RNA-Seq in MS

Line 246 – remove ‘firstly’ or add other points to the sentence.

R=Thank you for your suggestion. We have removed 'firstly' from line 246 to enhance the clarity of the sentence.

Lines 276 312 – same reference 

R=Thank you for your suggestion. We have made the modification in line 312 as suggested, using the reference from RÖHN, Gabriele et al. "ACTB and SDHA are suitable endogenous reference genes for gene expression studies in human astrocytomas using quantitative RT-PCR. Technology in cancer research treatment, v. 17, p. 1533033818802318, 2018."

Reviewer #1: Comments on the app

• In the Shiny app, on the “Gene Reference Analysis” page – please provide some extra written context to help with interpretation. For example, that the numbers in the topmost table are p-values, with significant p-vals in red. 

For the Normfinder Analysis, a brief description of how to interpret results to select the optimal combination of reference genes would be helpful (either on this page or the tutorial page). For example, what range of stability is acceptable for a gene to be retained as a reference, and in what range should one consider removing that gene before proceeding with gene expression analysis. 

R=Thank you. We added the following text at the Gene Reference Analysis page:

Gene Reference by group

The first line of the table is the p-value of the Kruskall-Wallis test for each reference gene.

The other lines are p-values of the Wilcoxon-Mann-Whitney test between each condition/group.

The last column is the test p-value of the reference genes mean.

Red values indicate p-values 0.05. It is desirable that reference genes and/or the reference genes mean do not vary significantly among groups/conditions.

Gene Reference Descriptive Statistics

It is preferable for reference genes not to vary across different conditions or groups.

Additionally, it is desirable for reference genes to exhibit low variances,

enhancing the ability to discern small yet significant differences in the target.

If a reference gene is varying across group/condition and/or exhibits high variances and/or has low stability, consider removing it from the analysis.

Normfinder Analysis

Normfinder is a method to evaluate reference gene stability using the variation of the gene expression.

The lower the stability value, the more stable the gene is considered.

Indeed, the ReadMe/Tutorial should provide a tutorial using the sample data for how to correctly analyse this dataset – what parameters provide evidence that a reference gene should be included/excluded, how this affects the interpretation of gene expression results – 

e.g. the tutorial could show how not excluding a reference gene with differential expression results in an incorrect conclusion on the differential analysis page. While the MS does go some way to assist in interpretation of results, the authors should assume that some users may access the app without reading the paper.

R= Thank you for your suggestion. At the tutorial section we added an extensive explanation of the tool, using the new example data set. We included data of a reference gene that should clearly be removed from the analysis, and showed its impact at the differential analysis.

• The same comment applies to the Differential Analysis page. Please provide some context to assist in interpreting results. The MS (line 98) states that fold-change results are provided on this page, but the word “fold” is not used on this page at all, making it potentially difficult to interpret/understand the provided results (especially to researchers new to the field). Also suggest renaming “Differential Analysis” to “Differential Expression Analysis” or similar (to make clear that this page is about analysing the actual results of a gene expression experiment, after prior reference gene selection.

R=Thank you. We changed the section name, as suggested. Included the topic "fold-change" with succinct explanation of its meaning at the tutorial section.

• On the Gene Reference samples page – it would be useful to have functionality to add outliers back into the analysis if the user chooses, so that the impact of removal and retention of these samples can more easily be explored. Also suggest adding some additional wording about what the two options for the radio buttons mean – this is clearer in the MS (lines 126-127) then in the software itself.

R=Thank you. We added a button to restore removed outliers, as suggested, and explain the impact of the two radio button options at the tutorial section.

• On the Gene Reference Analysis page – Have you considered adding an analysis to determine the optimal number and/or combination of reference genes? Many of the algorithms do this. Using the intragroup variance would be a good place to start. As stated in the original Normfinder paper: “Intragroup variance estimates provide a natural way of identifying the number of genes to include. The optimal number of genes is reached when addition of a further gene leads to a negligible reduction in the average of the gene variance estimates.”

Thank you for the suggestion. While this analysis wasn't initially included in the design, we can incorporate it in future versions, along with other features based on user feedback. 

• On the Gene Reference Analysis page – please provide a key for abbreviations (IGroup = intergroup, SD = standard deviation etc)

Thank you for your suggestion. We added in this section the respectives key words for abbreviations. 

• On the “Differential Analysis” page – it might be useful to offer an option to download the ggplot script as well as the PDF of the figure. This would allow users to edit the figure for publication. Also an option to download the tables. Figure and table download options on the Data Summary, Gene Reference Samples and Gene Reference Analysis pages would also be useful, for example so outputs can be saved in electronic notebooks. These are just suggestions though!

R= In the Differential Expression Analysis section, we've introduced a button that allows users to download all analysis results. This report includes the script employed to generate the Expression Boxplot. We appreciate your suggestion; however, we believe the steps involved in analyzing the reference genes are intermediate to obtaining the final result. Therefore, we have decided, for the time being, not to include downloads for these intermediate reports.

• I found the timing out somewhat annoying. Would it be possible to, for e.g., add a popup warning box along the lines of “you are about to be disconnected, do you wish to continue this session?". Or at least note on the tutorial page the amount of inactivity time a user has before they are disconnected from the server. Continually having to reupload the data was quite tedious.

R=Thank you for your suggestion. As far as we know, the time out is controlled by the shiny server. We may give more time to each user, however we don't have any more management over it. 

• It would be helpful if the sample data file had a larger set of candidate reference genes, with some that obviously show differential expression and should therefore be excluded, and others that are very stable and should be included. In the currently provided dataset, interpretation is a little ambiguous (each ref gene is stable for some comparisons but not others. While this likely mimics real-world data much of the time, for the purposes of a demo dataset, more clear-cut interpretation would be useful). The file should also include more (and some very obvious) sample outliers.

R=Thank you for your suggestion. We update the sample dataset, with clearer examples. And used it at the tutorial section for users to follow it step-by-step, explaining each step of the analysis.

Reviewer #2: Comments on the MS

1. Data Origin and Description:

- The manuscript lacks a clear explanation of the origin of the test data used in the analysis. The phrase "To illustrate the usefulness of the EndoGeneAnalyzer, we used the RT-qPCR data found at https://npobioinfo.shinyapps.io/endogeneanalyzer/" is insufficient. You need to provide a comprehensive description of how you obtained the RT-qPCR data, including data sources, collection methods, and any relevant details.

R= We appreciate your detailed observations. In response to your concern regarding the origin of the test data used in the analysis, we would like to clarify that the data were obtained from an unpublished experiment conducted in our laboratory. We aimed to ensure the integrity and authenticity of the data, reflecting a real-world context. It is pertinent to note that, while the data originate from a genuine experiment, we made alterations to the names of the target genes and the groups/conditions to preserve confidentiality and avoid any potential conflicts of interest.

2. Methodological Issues:

- There are issues with the method for identifying outliers that are unclear. It's vital to provide a detailed and transparent explanation of the outlier detection process.

Outliers were identified as samples with a mean Cq two standard deviations away from their respective group in a specific gene, or from the mean of all reference genes (based on the user's selection of 'All outliers' or 'Reference Mean'). This information has been incorporated into the tool tutorial.

- The use of boxplots combined with dots for data visualization is questioned. Authors should justify this choice or consider alternative visualization methods.

A combination of boxplot with scatter plots is intriguing for observing the distribution of samples, which may be obscured by outliers and small sample-sized groups. However, we acknowledge that there are various ways to visualize this data. Therefore, we have included a functionality that allows users to choose the best plot type, including boxplot + dispersion, boxplot, scatter plot, and violin plot. Additionally, the tool generates a report, providing users access to tables and scripts for the plots, enabling them to create their own visualizations

- The origin of values in the table "Reference of genes per group" / "Gene Reference by Group" and the color-coding is not adequately explained. Clarify what the values represent and the rules for color-coding.

R= Red values indicate p-values 0.05. We added a text in this section informing it. We also explained it at the tutorial section.

- The paper lacks a clear rationale for the choice of statistical tests for assessing the stability of reference genes. Additionally, it does not compare the proposed tool with existing reference gene selection methods, such as NormFinder, geNorm, BestKeeper, RefGenes, or RefFinder, to establish its superiority.

R=Statistical tests were conducted to assess the variation of reference genes across different groups/conditions. This is crucial as, when calculating Delta Cq, any variability in reference genes can be transferred to target genes, potentially leading to misinterpretations in the differential analysis. This serves as an extra measure of stability, complementing the stability assessment provided by NormFinder. Furthermore, we integrated an outlier analysis, distinguishing our tool from others that evaluate reference genes but do not include this additional analytical step.

- The reasons for conducting differential analysis and normality tests are unclear. Authors need to provide a more comprehensive description of how these steps aid in the selection of reference genes.

R=The differential expression analysis is an additional step that is not directly related to the evaluation of reference genes. Nevertheless, considering that all reference genes are already integrated into the tool, we believe it would enhance the tool's appeal and utility if the analysis of differential expression for target genes were also included.

3. Conclusion:

- The conclusion is notably brief and fails to adequately summarize the findings. It lacks an overall assessment of the tool's effectiveness and significance.

R=Thank you for your suggestion. In our revised manuscript, we will ensure to provide a more comprehensive summary of the findings, including a thorough assessment of the tool's effectiveness and significance.

4. Inconsistencies and Confusing Statements:

- The reference to "Postgraduate program in Biotechnology, Federal University of Pará, Belém, Pará, Brazil" as the institution appears unusual and should be clarified. 

R= Thank you for your observation, and we understand the need to clarify the reference to the "Postgraduate program in Biotechnology, Federal University of Pará, Belém, Pará, Brazil" in our manuscript. We recognize that this reference may appear unusual, and we agree to remove the citation to avoid confusion.

- Several sentences throughout the manuscript are confusing and should be rephrased for clarity such as lines 31-34 and 54-57. 

R= We appreciate your feedback, and we have made several improvements to the text based on your suggestions to enhance clarity.

- The use of "Ct (threshold cycle)" is inconsistent with MIQE guidelines, which recommend "Cq (quantification cycle)" values. 

R= We have changed Ct to Cq as proposed to MIQE guidelines.

Bustin SA, Benes V, Garson JA, Hellemans J, Huggett J, Kubista M, Mueller R, Nolan T, Pfaffl MW, Shipley GL, Vandesompele J, Wittwer CT. The MIQE guidelines: minimum information for publication of quantitative real-time PCR experiments. Clin Chem. 2009 Apr;55(4):611-22

- Address the ambiguity in the statement "ΔCt greater or less than 2" (Line 124) for better clarity.

R=Thank you for your observation. We would like to inform you that the statement "(ΔCt greater or less than 2)" on Line 124 has been corrected to "(∆Cq |2| standard deviations)" for improved clarity. 

- Correct the statement "(p-value in reference genes between the studied groups or conditions" (Line 137) for clarity. 

R=Thank you for your observation. We would like to inform you that the statement "(p-value in reference genes between the studied groups or conditions)" on Line 137 has been corrected to "(p-value in reference genes between the studied groups or conditions)" for improved clarity. The necessary modifications have been made in the revised version of the manuscript.

5. References:

- Ensure that references are correctly linked to their respective tools (line 60), and provide an accurate reference list.

R=Thank you for your feedback. We have made the suggested modification by citing the reference directly after each algorithm, as opposed to grouping all references at the end of the sentence.

- Correct punctuation and grammar in the text, including missing commas.

6. Figures:

- Figures 2, 3, 4, and 5 do not correspond with the current state of the tool, leading to confusion. Update these figures to accurately represent the tool.

R= Thank you for your suggestion. The figures 2, 3, 4 and 5 have been updated to reflect the current status of the tool.

- Figure 1 is misleading due to the use of dotted lines. Reconfigure the workflow diagram to eliminate confusion and better summarize the steps for users.

R= Thank you for your suggestion. The figure 1 have been updated to reflect the current status of the tool.

Reviewer #2: Comments on the app

7. Tool Inconsistencies:

- Address inconsistencies in the tool's interface and the manuscript regarding the naming of sections and options.

R= Thank you for your observation. We have carefully reviewed and corrected the inconsistencies pointed out between the tool's interface and the manuscript, specifically regarding the naming of sections and options.

- Resolve issues with the "Update Target Gene" button not functioning correctly.

R= If feasible, we kindly request the reviewer to provide more specific feedback. We thoroughly examined the "Update Target Gene" button and did not identify any malfunctions. However, its functionality may lack interaction as it does not provide feedback to the user regarding the selection of target genes (although this can be observed in the following sample data in "target" column). To address this, we have developed a comprehensive tutorial, which has been added to the Tutorial Section. This tutorial explains and illustrates the usage of the "Update Target Gene" button.

- Clarify the purpose and effects of the "Select one or more Genes" option in the tool.

R= This checkbox is used to remove Reference Genes from the analysis. To address this, we have developed a comprehensive tutorial, which has been added to the Tutorial Section. This tutorial explains and illustrates the usage of the "Select one or more Genes" checkbox. 

- Address inconsistencies in the tool's interface and the manuscript regarding the naming of sections and options.

R= Thank you for your observation. We have carefully reviewed and corrected the inconsistencies pointed out between the tool's interface and the manuscript, specifically regarding the naming of sections and options.

- Resolve issues with the "Update Target Gene" button not functioning correctly.

R= If feasible, we kindly request the reviewer to provide more specific feedback. We thoroughly examined the "Update Target Gene" button and did not identify any malfunctions. However, its functionality may lack interaction as it does not provide feedback to the user regarding the selection of target genes (although this can be observed in the following sample data in "target" column). To address this, we have developed a comprehensive tutorial, which has been added to the Tutorial Section. This tutorial explains and illustrates the usage of the "Update Target Gene" button.

- Clarify the purpose and effects of the "Select one or more Genes" option in the tool.

R=We added the following text to the tool: "Gene Reference Descriptive Statistics

It is preferable for reference genes not to vary across different conditions or groups.

Additionally, it is desirable for reference genes to exhibit low variances,

enhancing the ability to discern small yet significant differences in the target.

If a reference gene is varying across group/condition and/or exhibits high variances and/or has low stability, consider removing it from the analysis.

- Make sure that the results in the NormFinder analysis are presented in a clear and understandable manner.

R= In the NormFinder section, we have included a brief introduction to the method: "NormFinder is a reliable method for evaluating the stability of reference genes by quantifying the variation in gene expression. The lower the stability value, the more stable the gene is considered."

Furthermore, we have developed a comprehensive tutorial that guides users on how to effectively utilize NormFinder and other analysis techniques to select reference genes more accurately. This tutorial will assist users in making informed decisions and improving the reliability of their gene expression studies.

- Add titles to all sections in the Differential Analysis tab for clarity.

R= We added the respective titles, as suggested. .

8. Data Saving and Reporting:

- Consider implementing a feature to save the results of all analyses.

R=Thank you for the suggestion. While we acknowledge that this could be a valuable feature, the current development and hosting structure of EndoGeneAnalyzer does not support the creation and management of user accounts. It is possible that we may explore implementing this feature in the future.

- Explore the possibility of generating downloadable reports for users, which can enhance the utility of the tool.

R= In the Differential Expression Analysis section, we've introduced a button that allows users to download a report with the analysis results. Thank you for the suggestion.

---

## [Decision Letter · Decision Letter 1]

21 Dec 2023

PONE-D-23-21180R1EndoGeneAnalyzer: a tool for selection and validation of reference genesPLOS ONE

Dear Dr. TEIXEIRA,

Thank you for submitting your manuscript to PLOS ONE. After careful consideration, we feel that it has merit but does not fully meet PLOS ONE’s publication criteria as it currently stands. Therefore, we invite you to submit a revised version of the manuscript that addresses the points raised during the review process.

We look forward to receiving your revised manuscript.

Kind regards,

Karel Sedlar, Ph.D.

Academic Editor

PLOS ONE

Journal Requirements:

**Additional Editor Comments:**

While you have already done substantial improvements to both, manuscript and your tool, there is still need for additional improvement as both reviewers raised numerous, yet only minor, concerns. Please do no prepare your revision in hurry. I am aware that the reviewing process is quite lengthy for your paper but the manuscript must meet publication criteria before it can be accepted. Please consider detailed proofreading of your manuscript and try to eliminate any other mistakes. The concerns that reviewers raised are mostly formal, so I believe you should be able to correct it quite easily.

Reviewers' comments:

Reviewer's Responses to Questions

**Comments to the Author**

1. If the authors have adequately addressed your comments raised in a previous round of review and you feel that this manuscript is now acceptable for publication, you may indicate that here to bypass the “Comments to the Author” section, enter your conflict of interest statement in the “Confidential to Editor” section, and submit your "Accept" recommendation.

Reviewer #1: (No Response)

Reviewer #2: (No Response)

2. Is the manuscript technically sound, and do the data support the conclusions?

Reviewer #1: Yes

Reviewer #2: Yes

3. Has the statistical analysis been performed appropriately and rigorously? 

Reviewer #1: Yes

Reviewer #2: Yes

4. Have the authors made all data underlying the findings in their manuscript fully available?

Reviewer #1: Yes

Reviewer #2: Yes

5. Is the manuscript presented in an intelligible fashion and written in standard English?

Reviewer #1: No

Reviewer #2: No

6. Review Comments to the Author

Reviewer #1: Well done to the authors for addressing my comments, I think the improvements to the app will make it much easier to use. My remaining comments are largely minor and can be easily addressed.

Comments (all line numbers relate to the tracked changes version of the MS)

Line 31 – change to ‘tool to assist’

Line 32 – lower case Q

Line 46 – remove The

Line 63 – only as it pertains to gene expression studies. RT-qPCR is used for many other purposes as well.

Lines 65-67 – This sentence should not be a paragraph, instead, make it the first sentence of the previous paragraph.

Lines 72-74 – cite Chapman and Waldenström for this point (stability of ref genes within study context)

Line 206 – remove second ‘the’ on this line

Some of the wording in the tutorial section of the app needs proof-reading (perhaps by a native English speaker) for clarity.

Have you considered having an option for Bonferroni (or other method) correction for multiple comparisons in the Differential Expression Analysis?

Reviewer #2: The revised manuscript and corresponding application have undergone significant improvements; however, some of my previous concerns were not fully addressed.

Comments on the manuscript:

I appreciate the creation of a comprehensive tutorial page for your tool. Nevertheless, some information is missing in the manuscript. The method for outliers removal, a previous concern of mine, is now clearly described in the tutorial page but not adequately addressed in the manuscript (lines 143-144). Additionally, the new functionality of restoring outliers is not mentioned in the manuscript.

While the authors provided a conclusion for their manuscript, the sentences are overly complicated, hindering understanding, and contain grammar mistakes. The conclusion should also mention the differential expression analysis as one of the additional features of this tool. Please delete the last sentence.

Regarding the grammar and flow of the text, I recommend having a native speaker review and edit the whole manuscript.

Comments on the figures:

Fig 1: Correct the name of third step from: "Gene Reference Sample" to "Gene Reference Samples".

Fig 2: Check the description of this figure in the manuscript. Change "Ct" to "Cq".

Fig 4: The values in the figure are currently presented only in black and white, despite the description referencing values highlighted in red. To ensure accuracy, update the figure to include the specified coloration. Additionally, align the numbering format consistently between the figure and its description (e.g., use 1, 2, 3 in both). Furthermore, incorporate the respective p-values associated with the color change into the figure description. This ensures that the significance of the color variation is appropriately communicated in the context of the data presented.

Other minor comments:

Lines 32 + 41: According to the MIQE guidelines (Nomenclature 1.1) RT-qPCR stands for "reverse transcription quantitative real-time polymerase chain reaction".

Line 55: Explain the meaning of Cq here.

Lines 58 – 61: Check the grammar and clarity of this sentence. Firstly, correct "data normalization process to expression comparison" to "data normalization process for expression comparison". Secondly, consider making the last addition a standalone sentence: "The normalization of data using reference genes involves correcting errors that arise from the initial concentration of RNA/cDNA. "

Line 67: Check references on this line. Either put respective reference after mentioned gene or put all references at the end of the sentence into single brackets.

Line 128: Delete comma in this sentence.

Line 150: Correct "Reference of genes by group" to "Gene Reference by group" to be consistent with the tool.

Lines 163 + 165: Check equations, especially the summation symbols and remove word "group" from the summation symbol, it is redundant.

Line 164: Remove this line.

Line 200: Change "tutorial" to "Tutorial".

Line 226: Change "meanRef" to "MeanRef".

Line 269: Check references on this line. Put all references at the end of the sentence into single brackets.

Comments on the App:

Regarding the "Update Target Gene": I was testing your tool and I have selected several genes and pressed the "Update Target Gene" button. After that I tried to unselect any targeted genes and press this button again, but nothing has changed and targets stayed the same. I do not think this a desirable feature.

Regarding this concern "Clarify the purpose and effects of the "Select one or more Genes" option in the tool.": Although the tutorial and the tool provide an explanation now, I would recommend to change the title "Select one or more Genes" to "Selected Reference Genes" to be more intuitive.

On the "Gene Reference Analysis" tab change all occurrences of "Normfinder" to "NormFinder".

On the "Tutorial" tab:

Point 1: Change "meanCT" to "mean Cq".

Point 2,3 and 5: Corresponding figures have on y-axis "meanCT", change it to "meanCq".

Point 5.2: Here, you mention "2^-ΔCq" and in the manuscript "2-ΔΔCq" formula.

Please correct the Tutorial page for consistency and check grammar.

Finally, let me make two more suggestions for the tool.

Regarding the "Gene Reference Samples" tab: I would recommend to add a short description to outlier removal methods, so user can read the description here and decide which method to use. I really like the added description for the "Gene Reference Analysis" tab, which made the analysis easier. I believe that this tab should be treated in the same way and this addition would make it more user-friendly.

Create a new tab in the tool called "About". There, provide your contact information for the users as you have also added a warning to the upload page saying "An error has occurred. Check your logs or contact the app author for clarification." but contact information is missing. I would also recommend to provide a link to GitHub page with your code and keep your GitHub page with codes updated. Users can leave feedback there, if they run into some problems, or if they request a new feature to be added.

In summary, commendable progress has been made in refining the manuscript and its corresponding application; however, there are still areas that require attention. Focusing on specific concerns, particularly in enhancing clarity, consistency, and functionality, will undoubtedly contribute to the continuous improvement and effectiveness of the tool. The commitment to elevate the overall quality and user experience of both the manuscript and the application is evident, and I look forward to seeing the implementation of these valuable improvements for the benefit of users and the advancement of your tool within the scientific community.

7. PLOS authors have the option to publish the peer review history of their article (what does this mean?). If published, this will include your full peer review and any attached files.

Reviewer #1: No

Reviewer #2: No

---

## [Author Response · Author response to Decision Letter 1]

15 Feb 2024

RESPONSE LETTER TO THE REVIEWERS

Dear reviewer, my co-authors and I would like to thank you for the suggestions made during this high-quality review and then we present the answer to the questions.

We inform that with the reviews and suggestions, we were able to improve the idea presented by our work and we appreciate the opportunity. We hope this review has left the article suitable for publication in this high-impact and prestigious journal.

Kind Regards.

Reviewer #1: Well done to the authors for addressing my comments, I think the improvements to the app will make it much easier to use. My remaining comments are largely minor and can be easily addressed.

Comments (all line numbers relate to the tracked changes version of the MS)

Line 31 – change to ‘tool to assist’

R=Thank you for your suggestion! The change to 'tool to assist' has been implemented.

Line 32 – lower case Q

R=Thank you for your suggestion! The change to lower case 'Q' on Line 32 has been implemented

Line 46 – remove The

R= Thank you for your suggestion! We remove ‘The’ on Line 46

Line 63 – only as it pertains to gene expression studies. RT-qPCR is used for many other purposes as well.

R= Thanks for your suggestion! We have clarified the sentence in Line 63 to explicitly state that in the study RT-qPCR refers only to gene expression studies.

Lines 65-67 – This sentence should not be a paragraph, instead, make it the first sentence of the previous paragraph.

R=Thank you for your input! We have reorganized the text as per your suggestion. The content from Lines 65-67 is now the first sentence of the preceding paragraph.

Lines 72-74 – cite Chapman and Waldenström for this point (stability of ref genes within study context)

R=Thank you for your suggestion! We cite Chapman and Waldenström on lines 72-74

Line 206 – remove second ‘the’ on this line

R= Thank you for your suggestion! We remove second ‘the’ on Line 206

Some of the wording in the tutorial section of the app needs proof-reading (perhaps by a native English speaker) for clarity.

R= Thank you for pointing it out, we reviewed the tutorial section wording.

Have you considered having an option for Bonferroni (or other method) correction for multiple comparisons in the Differential Expression Analysis?

R=After ANOVA, Tukey's post hoc test is used for multiple comparisons and already provides the adjusted p-value. Similarly, following Kruskal-Wallis, the Dunn post hoc test is applied for multiple comparisons, and the p-values are adjusted using the Benjamini-Hochberg method. We added this information at the tutorial section.

Reviewer #2: The revised manuscript and corresponding application have undergone significant improvements; however, some of my previous concerns were not fully addressed.

Comments on the manuscript:

I appreciate the creation of a comprehensive tutorial page for your tool. Nevertheless, some information is missing in the manuscript. The method for outliers removal, a previous concern of mine, is now clearly described in the tutorial page but not adequately addressed in the manuscript (lines 143-144). Additionally, the new functionality of restoring outliers is not mentioned in the manuscript.

R= Thank you for your feedback. We appreciate your acknowledgment of the comprehensive tutorial and note your concerns. We will enhance the manuscript by providing a more detailed explanation of the outliers removal method (lines 143-144) and include information about the new functionality for restoring outliers. 

While the authors provided a conclusion for their manuscript, the sentences are overly complicated, hindering understanding, and contain grammar mistakes. The conclusion should also mention the differential expression analysis as one of the additional features of this tool. Please delete the last sentence.

Regarding the grammar and flow of the text, I recommend having a native speaker review and edit the whole manuscript.

R= We've made the changes as requested, and the manuscript has been sent for translation by native speakers. Regarding the conclusion, the sentences were overly complex, making it difficult to understand, and there were grammar mistakes. We've also added a mention of the differential expression analysis as one of the additional features of the tool. However, we've removed the last sentence. 

Comments on the figures:

Fig 1: Correct the name of third step from: "Gene Reference Sample" to "Gene Reference Samples".

R= Thank you for your feedback. We have made the correction in Fig 1 by updating the name of the third step from 'Gene Reference Sample' to 'Gene Reference Samples.

Fig 2: Check the description of this figure in the manuscript. Change "Ct" to "Cq".

R= Thank you for your observation. We have revised the description of Fig 2 in the manuscript, changing 'Ct' to 'Cq' as per your suggestion."

Fig 4: The values in the figure are currently presented only in black and white, despite the description referencing values highlighted in red. To ensure accuracy, update the figure to include the specified coloration. Additionally, align the numbering format consistently between the figure and its description (e.g., use 1, 2, 3 in both). Furthermore, incorporate the respective p-values associated with the color change into the figure description. This ensures that the significance of the color variation is appropriately communicated in the context of the data presented.

R= Thank you for your observation. For Figure 4, we've updated the coloration to include the specified red highlighting to match the description accurately. 

Other minor comments:

Lines 32 + 41: According to the MIQE guidelines (Nomenclature 1.1) RT-qPCR stands for "reverse transcription quantitative real-time polymerase chain reaction".

R= Thank you for your suggestion. We have implemented the change, and the text now correctly states: 'RT-qPCR stands for reverse transcription quantitative real-time polymerase chain reaction.

Line 55: Explain the meaning of Cq here.

R=Thank you for your feedback. We have addressed the request on Line 55 and provided an explanation for the meaning of Cq.

Lines 58 – 61: Check the grammar and clarity of this sentence. Firstly, correct "data normalization process to expression comparison" to "data normalization process for expression comparison". Secondly, consider making the last addition a standalone sentence: "The normalization of data using reference genes involves correcting errors that arise from the initial concentration of RNA/cDNA.

R= Thank you for your suggestions. We have addressed the concerns in Lines 58-61. Firstly, we corrected "data normalization process to expression comparison" to "data normalization process for expression comparison." Additionally, we separated the last addition into a standalone sentence, which now reads: 'The normalization of data using reference genes involves correcting errors that arise from the initial concentration of RNA/cDNA.'

Line 67: Check references on this line. Either put respective reference after mentioned gene or put all references at the end of the sentence into single brackets.

R= Thank you for your guidance. We have revised Line 67 by placing all references at the end of the sentence within single brackets.

Line 128: Delete comma in this sentence.

R= Thank you for your suggestion. The comma has been deleted in this sentence.

Line 150: Correct "Reference of genes by group" to "Gene Reference by group" to be consistent with the tool.

R=Thank you for your feedback. The change "Reference of genes by group" has been corrected to "Gene Reference by group" for consistency with the tool.

Lines 163 + 165: Check equations, especially the summation symbols and remove word "group" from the summation symbol, it is redundant.

R=Thank you for your feedback. Equations have been checked, and the word "group" has been removed from the summation symbol as it was redundant.

Line 164: Remove this line.

R=Thank you for your suggestion. This line has been removed.

Line 200: Change "tutorial" to "Tutorial".

R=Thank you for your suggestion. We change "tutorial" to "Tutorial."

Line 226: Change "meanRef" to "MeanRef".

R=Thank you for your suggestion. We change "meanRef" to "MeanRef."

Line 269: Check references on this line. Put all references at the end of the sentence into single brackets.

R= Thank you for your suggestion. References on this line have been checked, and all references at the end of the sentence are now in single brackets.

Comments on the App:

Regarding the "Update Target Gene": I was testing your tool and I have selected several genes and pressed the "Update Target Gene" button. After that I tried to unselect any targeted genes and press this button again, but nothing has changed and targets stayed the same. I do not think this a desirable feature.

R= Thank you for your careful review. We hadn't noticed this misbehavior. The issue has been fixed.

Regarding this concern "Clarify the purpose and effects of the "Select one or more Genes" option in the tool.": Although the tutorial and the tool provide an explanation now, I would recommend to change the title "Select one or more Genes" to "Selected Reference Genes" to be more intuitive.

R= Suggestion accepted. Thank you.

On the "Gene Reference Analysis" tab change all occurrences of "Normfinder" to "NormFinder".

R= Suggestion accepted. Thank you.

On the "Tutorial" tab:

Point 1: Change "meanCT" to "mean Cq". 

R= Thank you for your careful review.Suggestion accepted. 

Point 2,3 and 5: Corresponding figures have on y-axis "meanCT", change it to "meanCq". 

R= Thank you for your careful review.Suggestion accepted. 

Point 5.2: Here, you mention "2^-ΔCq" and in the manuscript "2-ΔΔCq" formula. 

R= The term 2^-ΔCq is employed for estimating expression levels, while 2^-ΔΔCq is utilized for estimating fold-change. In this section, where we elucidate the estimation of expression levels, the formula was applied within the appropriate context.

Please correct the Tutorial page for consistency and check grammar. 

R=Thank you for your careful evaluation. Suggestion accepted. The tool page and manuscript have been revised.

Finally, let me make two more suggestions for the tool.

Regarding the "Gene Reference Samples" tab: I would recommend to add a short description to outlier removal methods, so user can read the description here and decide which method to use. I really like the added description for the "Gene Reference Analysis" tab, which made the analysis easier. I believe that this tab should be treated in the same way and this addition would make it more user-friendly.

R= Thank you for your careful review.Suggestion accepted. 

Create a new tab in the tool called "About". There, provide your contact information for the users as you have also added a warning to the upload page saying "An error has occurred. Check your logs or contact the app author for clarification." but contact information is missing. I would also recommend to provide a link to GitHub page with your code and keep your GitHub page with codes updated. Users can leave feedback there, if they run into some problems, or if they request a new feature to be added.

R=We have implemented the "About" tab as recommended, incorporating a concise tool description, contact and support details, GitHub link, information about participants and funding partners. Additionally, upon acceptance, we will include the publication link.

In summary, commendable progress has been made in refining the manuscript and its corresponding application; however, there are still areas that require attention. Focusing on specific concerns, particularly in enhancing clarity, consistency, and functionality, will undoubtedly contribute to the continuous improvement and effectiveness of the tool. The commitment to elevate the overall quality and user experience of both the manuscript and the application is evident, and I look forward to seeing the implementation of these valuable improvements for the benefit of users and the advancement of your tool within the scientific community.

---

## [Editor Report · Decision Letter 2]

20 Feb 2024

EndoGeneAnalyzer: a tool for selection and validation of reference genes

PONE-D-23-21180R2

Dear Dr. TEIXEIRA,

We’re pleased to inform you that your manuscript has been judged scientifically suitable for publication and will be formally accepted for publication once it meets all outstanding technical requirements.

Kind regards,

Karel Sedlar, Ph.D.

Academic Editor

PLOS ONE

Additional Editor Comments (optional):

Congratulations! I believe your contribution will be of great use for the community. Special thanks also belongs to reviewers. I am once againg sorry it took so long to get them but it was not easy to get specialists in your specific field. Nevertheless, their thorough reviews helped to improve your manuscript and tool a lot.